# The antenna of far-red absorbing cyanobacteria increases both absorption and quantum efficiency of Photosystem II

Vincenzo Mascoli [1], Ahmad Farhan Bhatti[2], Luca Bersanini[1], Herbert van Amerongen [2,3] & Roberta Croce [1✉]

Cyanobacteria carry out photosynthetic light-energy conversion using phycobiliproteins for light harvesting and the chlorophyll-rich photosystems for photochemistry. While most cyanobacteria only absorb visible photons, some of them can acclimate to harvest far-red light (FRL, 700–800 nm) by integrating chlorophyll *f* and *d* in their photosystems and producing red-shifted allophycocyanin. Chlorophyll *f* insertion enables the photosystems to use FRL but slows down charge separation, reducing photosynthetic efficiency. Here we demonstrate with time-resolved fluorescence spectroscopy that on average charge separation in chlorophyll-*f*-containing Photosystem II becomes faster in the presence of red-shifted allophycocyanin antennas. This is different from all known photosynthetic systems, where additional light-harvesting complexes increase the overall absorption cross section but slow down charge separation. This remarkable property can be explained with the available structural and spectroscopic information. The unique design is probably important for these cyanobacteria to efficiently switch between visible and far-red light.

[1] Department of Physics and Astronomy and Institute for Lasers, Life and Biophotonics, Faculty of Sciences, Vrije Universiteit Amsterdam, Amsterdam, The Netherlands. [2] Laboratory of Biophysics, Wageningen University & Research, Wageningen, the Netherlands. [3] MicroSpectroscopy Research Facility, Wageningen University & Research, Wageningen, The Netherlands. ✉email: r.croce@vu.nl

The light-dependent reactions of oxygenic photosynthesis are carried out by large assemblies of proteins, pigments, and electron carriers, known as Photosystem I (PSI) and Photosystem II (PSII) supercomplexes. These supercomplexes consist of an outer antenna performing light harvesting and a chlorophyll-rich photosystem core. The latter collects the excitations formed in the antenna and uses their energy to power charge separation through a pigment cluster in the centrally-located reaction center (RC) complex[1]. While the photosystem cores, which are membrane-embedded, have a highly similar architecture in all oxygenic photoautotrophs, the antenna systems are more variable in both size and composition. This flexibility is required to cope with specific environmental conditions (such as the spectrum and/or intensity of light available)[2].

While plants and most algae employ membrane-bound light-harvesting complexes[3], cyanobacteria have water-soluble antennas, the phycobilisomes (PBSs)[4–6]. PBSs are relatively large complexes (typically several MDa) of phycobiliproteins containing covalently-bound chromophores named bilins, and associated linker proteins. Most cyanobacteria have a hemidiscoidal PBS consisting of a core composed of two, three, or five cylinders of allophycocyanin (APC) subunits, two of which are connected to the photosystem cores, and several rods of phycocyanin and, sometimes, phycoerythrin[7–9]. The downhill energy gradient between the peripheral rods and the APC core ensures directionality in the energy transfer towards the photosystems[10]. By binding up to several hundred pigments[11], the PBSs enormously increase the absorption cross section of the photosystems and expand the photosynthetically active spectrum to regions (commonly between 500 and 650 nm) where the absorption by the chlorophylls (Chls) is weak. For most photosynthetic systems, however, a general consequence of using a larger antenna is a longer time required for the excitations to reach the RCs and power photochemistry, resulting in a lower photochemical efficiency[12,13].

While most photoacclimation strategies pertain to the antenna, a group of cyanobacteria growing in deep-shaded environments can acclimate to harvest far-red light (FRL, 700–800 nm) by remodeling both their antenna and core photosystems[14]. Under visible light (such as white light, WL), these organisms produce only one type of Chl, Chl *a*, and absorb visible photons like "conventional" cyanobacteria. Under FRL, however, they become capable of harvesting less energetic quanta by synthesizing the red-shifted Chls *f* and *d*[14–16]. FRL-induced photoacclimation (FaRLiP) involves replacing the photosystems expressed in WL (WL-photosystems) with new photosystems containing FRL-specific paralog protein subunits. FRL-photosystems incorporate a small number of chlorophylls *f* and *d* allowing them to absorb up to 750 nm (FRL-PSII)[17–20], or even 800 nm (FRL-PSI)[17,19,21–23]. Under FRL, new APC paralog subunits are produced, which assemble into bicylindrical cores (FRL-BCs)[24,25]. While WL-PBSs absorb up to 670–680 nm, FRL-BCs contain a large number of red-shifted bilins absorbing above 700 nm[18,24,25]. The organization of phycobiliproteins in FaRLiP organisms is species-dependent. Some strains possess only FRL-BCs in FRL[25], while in others the peripheral rods of phycocyanin and phycoerythrin are associated with FRL-BCs[26]. However, in all strains analyzed including *Chroococcidiopsis thermalis*, the phycobilisomes produced in FRL (FRL-PBSs) are smaller than those assembled in WL[27]. Finally, besides producing FRL-photosystems and FRL-BCs, some strains were shown to maintain variable amounts of WL-photosystems and/or WL-PBSs after weeks of acclimation to FRL[18,19,24].

The use of red-shifted pigments represents a promising strategy for extending the photosynthetically active spectrum of other organisms, such as plants and algae. In order to achieve high biomass yields, however, the newly engineered photosynthetic units need to preserve high photochemical yields. In this respect, we have recently shown that the insertion of Chl *f* slows down photochemistry in both photosystems and significantly decreases the efficiency of FRL-PSII in comparison to that of WL-PSII[19]. This effect is ascribed to (i) slower charge separation by the RCs due to a red-shifted primary electron donor and (ii) less effective connectivity between the few FRL-absorbing Chls surrounded by a majority of Chls *a*. While the latter study investigated the two photosystems, little is known about the fate of the excitations harvested by their outer antenna, especially at physiological temperatures. This knowledge is particularly relevant because FRL-BCs absorb a substantial amount of FRL in the cells, implying that their performance has a considerable impact on the overall light-energy conversion. To shed more light on these aspects, we measured time-resolved fluorescence (TRF) of intact cells of the FaRLiP strain *Chroococcidiopsis thermalis* acclimated to FRL and compared these results with those from WL-acclimated cells. We performed experiments at two different excitation wavelengths to preferentially excite the Chls or the bilins, thereby investigating the fate of the energy absorbed by either the photosystems or the phycobiliproteins. These data were successively used to quantify the efficiency of light conversion after exciting different units.

## Results

**Steady-state spectroscopy.** In comparison with cells acclimated to WL (WL-cells), the cells acclimated to FRL (FRL-cells) display an additional absorption band above 700 nm (Fig. 1a).

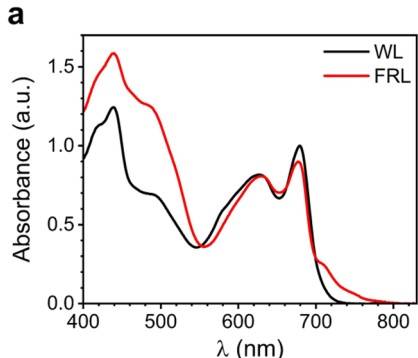

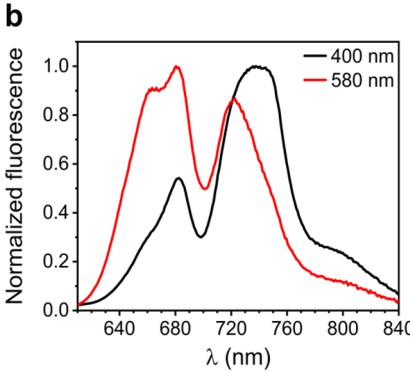

**Fig. 1 Steady-state spectra at room temperature of intact cells acclimated to WL or FRL. a** Absorption spectra of WL- and FRL-cells (normalized to the area in the region above 580 nm). **b** Fluorescence spectra of FRL-cells excited at 400 and 580 nm. The corresponding spectra of WL-cells are displayed in Supplementary Fig. 1.

This is caused by the integration of several Chls $f$ in the FRL-photosystems and by the synthesis of red-shifted APC forming FRL-PBS[14]. The Chl content of FRL-cells is dominated by Chl $a$ (~95%), whereas the remaining ~5% can mainly be ascribed to Chl $f$, with traces of Chl $d$[19]. The extra absorption of FRL-cells in the blue-green region ($\lambda < 530$ nm) has also been observed in other FaRLiP strains acclimated to FRL and ascribed to an increased carotenoid content[16,21,28].

The fluorescence spectra of FRL-cells show two distinct "bands" upon excitation of both Chls (400 nm) and bilins (580 nm) (Fig. 1b). The fluorescence below 700 nm is due to WL-PSII and WL-PBSs (see also Supplementary Fig. 1 for spectra of WL cells) that are maintained in FRL, in line with previous findings on another FaRLiP strain[18]. The fluorescence above 700 nm is dominated by FRL-PSII and the FRL-BCs[19], while the shoulder extending up to 800 nm stems from the extremely red-shifted Chls $f$ typical for FRL-PSI[17,19,23]. Upon 580-nm excitation, which is more selective for the bilins, the far-red band is more blue-shifted than upon 400-nm excitation due to a larger contribution from FRL-BCs. The strong dependency of far-red fluorescence on the excitation wavelength indicates that FRL-BCs and FRL-photosystems do not fully equilibrate before excited-state decay.

**Trapping of Chl and bilin excitations by FRL-PSII RCs.** To investigate the fate of different pigment excitations, the excited-state dynamics of intact cells were probed via TRF (Fig. 2 and Supplementary Figs. 2–9). Cells were excited both at low power,

when most PSII RCs are in the open state, and at high power in the presence of 3-(3,4-dichlorophenyl)-1,1-dimethylurea (DCMU), when PSII RCs are in the closed state, to separate the spectroscopic contribution of PSII from that of PSI and estimate the photochemical yield of PSII (see below). Use is made of the fact that the fluorescence decay of PSI remains essentially unaltered upon closure of the RCs whereas the average lifetime of PSII slows down considerably. Experiments were performed at two different excitation wavelengths, 400 nm (more selective for the photosystems and yielding results similar to our previous work[19]) and 577 nm (more selective for the phycobiliproteins). Due to the heterogeneity of the photosynthetic apparatus of FRL-cells, both WL-PBSs and FRL-PBSs are excited at 577 nm. Note that it is not possible to excite FRL-PBSs selectively, as their absorption maxima (around 650 nm and 710 nm)[24,25] overlap with the absorption bands of the Chls. However, an effective separation of the signatures from WL-PBSs and FRL-PBSs is possible from the analysis of the spectrally-resolved TRF data. Below we only show the main findings for FRL-cells, whereas the study of WL-cells and the results from global analysis of all TRF datasets can be found in the Supplementary Materials (see Supplementary Figs. 3–8).

For FRL-cells excited at 577 nm, the TRF traces detected at 730 nm (where both FRL-PBSs and FRL-PSII emit) and normalized to their maxima become substantially longer-lived when PSII RCs close (Fig. 2a). This indicates that FRL-PBSs, which are more selectively excited at 577 nm, transfer their excitations mainly to FRL-PSII and are able to power photochemistry. The corresponding

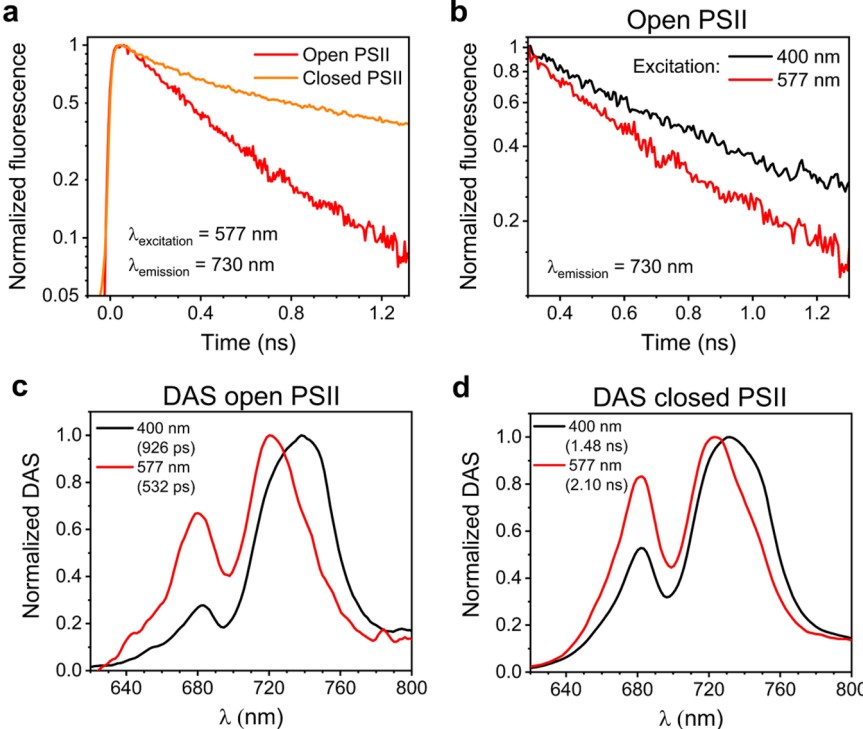

**Fig. 2 TRF of FRL-cells at room temperature. a** TRF traces (normalized to their maxima) were detected at 730 nm (where both FRL-BCs and FRL-PSII emit) upon 577-nm excitation with mostly open and closed PSII RCs. The presented traces were obtained by integrating the TRF data between 725 and 735 nm and were binned to a time-step of 6 ps. **b** TRF traces were detected at 730 nm upon 400-nm and 577-nm excitation with mostly open PSII RCs. The traces are normalized to their value around 300 ps, and the first 300 ps of fluorescence kinetics are excluded for clarity. Please note that, in **a** and **b**, the fluorescence intensity is shown on a logarithmic scale. **c** Normalized long-lived DAS from TRF measurements with open PSII RCs after 400-nm and 577-nm excitation (corresponding to the green DAS in Supplementary Figs. 6A and 7A). **d** Normalized long-lived DAS from TRF measurements with closed PSII RCs after 400-nm and 577-nm excitation (corresponding to the green DAS in Supplementary Fig. 6B and the magenta DAS in Supplementary Fig. 7B). The lifetimes of the DAS are indicated in parentheses in the legends. Note that, for the DAS in **c** and **d**, the signal below 700 nm stems mostly from the WL-PSII and WL-PBSs that are maintained in the FRL-cells, whereas that above 700 nm is emitted by FRL-BCs and FRL-PSII.

traces obtained upon 400-nm excitation, which is more selective for the photosystems, are shown in Supplementary Fig. 2. Note that, at earlier delays, the 400-nm-excited traces are dominated by the fast-decaying Chls *f* of FRL-PSI. As a result, the TRF traces of Fig. 2a cannot be simply juxtaposed to those in Supplementary Fig. 2 to compare the trapping rates of Chl *f* and bilin excitations by PSII RCs. For a qualitative comparison, we therefore overlaid the 400-nm and 577-nm excited traces in open state normalized at a delay of 300 ps (Fig. 2b). This delay is sufficiently longer than the timescale of energy transfer and trapping in FRL-PSI (< 200 ps, see also Supplementary Fig. 6)[19,29], implying that the signal detected at 730 nm after 300 ps stems mainly from the FRL-PBSs and FRL-PSII. When the 577-nm and 400-nm excited traces are normalized at this delay, the former trace is substantially shorter-lived. This is partly due to a faster decay component of PBS fluorescence (with a time constant of 161 ps as can be seen in Supplementary Fig. 7)), especially in the earliest period after time-point 300 ps. However, as shown by the different slopes of the traces in Fig. 2b, it is evident that upon 400 nm excitation the fluorescence decay is slower also at later times. This implies that trapping of FRL-PBSs excitations by FRL-PSII RCs is, on average, faster than trapping from the Chls *f*. This is also illustrated by the results of the global analysis of the fluorescence data. The lifetime of the longest-lived DAS (decay-associated spectrum) in the open state, which can be used to estimate the time required to achieve charge separation, decreases from ~930 ps upon 400-nm-excitation to ~530 ps upon 577-nm excitation (Fig. 2c; see Supplementary Figs. 6 and 7 for the full set of DAS). This confirms that energy trapping after FRL-PBS excitation is markedly faster than after Chl excitation. At the same time, the ~930-ps DAS after preferential Chl excitation and the ~530-ps DAS after bilin excitation do not overlap and their peaks are shifted by nearly 20 nm. This indicates that the former DAS stems mostly from Chls *f*, whereas the latter is enriched in red-shifted APC. A similar difference between the two excitations is observed for the longest-lived DAS when PSII RCs are closed (Fig. 2d). These results imply that the excitations in the FRL-BCs and those in the FRL-PSII antenna do not fully equilibrate, even after 1–2 ns.

Evidently, the association of FRL-PBSs with FRL-PSII in FRL-acclimated *Chroococcidiopsis thermalis* appears to be different from that of PBSs and PSII cores of previously studied cyanobacteria[30,31] and WL-acclimated *Chroococcidiopsis thermalis*. In the latter case, WL-PSII RCs trap the excitations formed in WL-PBSs more slowly than those formed on the chlorophylls of the WL-PSII core. Furthermore, excitations of the bilins of WL-PBSs and of the Chls *a* of WL-PSII equilibrate after few hundred picoseconds (Supplementary Fig. 3). It is important to realize that the 532-ps DAS in Fig. 2c, besides the 720 nm peak, reflecting trapping by FRL-PSII, also has a 680-nm peak due to WL-PSII (see also the 503 ps component in Supplementary Fig. 5a). We explain the fact that FRL-PSII and WL-PSII have very similar trapping times upon 577-nm excitation despite the (far) slower trapping in the FRL-PSII core itself, by the smaller size of the FRL-PBSs (see also Introduction), which may well be constituted by FRL-BCs only.

**Efficiency of PSII photochemistry upon different pigment excitations**. The TRF data presented above can be integrated to obtain steady-state fluorescence spectra in the open ($F_o$) and closed ($F_m$) states to calculate the variable fluorescence ($F_v = F_m - F_o$) at various emission wavelengths (Fig. 3). Variable fluorescence only stems from the pigments transferring to PSII RCs. The ratio $F_v/F_m$ in the spectral regions where PSII emission is maximal can be used to estimate the quantum yield of PSII photochemistry, $\Phi_{PSII}$ (see Supplementary Fig. 10 for details on the validity of this approximation), which reflects the efficiency of

charge separation in the PSII RCs after light absorption by the Chls in PSII and the bilins of the PBSs.

At both excitation wavelengths (400 and 577 nm), the variable fluorescence spectrum of WL-cells peaks at ~680 nm and stems from WL-PSII (Fig. 3a, b), with a shoulder at shorter wavelengths due to the WL-PBS antenna. Upon 400-nm excitation, $F_v/F_m$ at 680 nm is nearly 0.6 (Fig. 3c, black). This value of $\Phi_{PSII}$ is likely underestimated because both the excitation and emission wavelengths are not entirely selective for the Chls of WL-PSII. The maximal $F_v/F_m$ drops to ~0.4 upon 577-nm excitation (Fig. 3c, red), which is likely the result of two simultaneous effects: (i) not all WL-PBSs transfer energy to WL-PSII and (ii) trapping by WL-PSII after PBS excitation is slower than trapping after direct WL-PSII excitation due to the additional time required for the energy migration from the WL-PBS to WL-PSII. Consequently, WL-PBSs increase the antenna size of WL-PSII at the expense of some photochemical efficiency.

The variable fluorescence of FRL-cells shows two distinct peaks (Fig. 3d, e): one around 680 nm, which we ascribe to WL-PSII and its WL-PBS antenna, and one above 700 nm due to FRL-PSII and its FRL-PBS antenna. The $F_v$ spectrum in the far-red region upon 400-nm excitation (Fig. 3d, blue) is clearly blue-shifted with respect to both $F_o$ and $F_m$ (Fig. 3d, black and red) and also relative to the spectrum of isolated FRL-PSII[19]. This indicates that trapping by FRL-PSII is more efficient at shorter wavelengths, where the contribution from FRL-PBSs is larger. $F_v/F_m$ is therefore 0.50 at 720 nm (Fig. 3f, black), it drops to about 0.44 at 740–750 nm, where fluorescence from the Chls *f* of FRL-PSII is maximal, and decreases even more at longer wavelengths, where FRL-PSI fluorescence dominates. Upon 577-nm excitation, as a consequence of the larger contribution from FRL-PBSs, $F_o$, $F_m$, and $F_v$ all peak around 720 nm in the far-red region (Fig. 3e). In addition, $\Phi_{PSII}$ at $\lambda > 700$ nm is substantially higher than after 400-nm excitation, rising to a maximum of 0.68 (Fig. 3f). All these results imply that FRL-PSII photochemistry after FRL-PBS excitation is more efficient than after direct Chl excitation, which is in contrast with what is normally observed for the super-complexes of WL-PBS and WL-PSII of cyanobacteria[32] (*cfr*. Fig. 3c, f). The main cause of this somewhat counter-intuitive finding is that trapping by FRL-PSII RCs after excitation of FRL-PBSs (requiring up to 500 ps) is remarkably faster—on average—than after Chl excitation (requiring up to 900 ps).

## Discussion

In WL-cells, the PSII RCs trap excitations formed in the adjacent PBSs more slowly (and, therefore, less efficiently) than the excitations formed directly on the Chls. Furthermore, excitations in the WL-PBSs and WL-PSII equilibrate to a large extent within hundreds of picoseconds. These results are in line with what is commonly observed in cyanobacteria. Conversely, FRL-PBSs can deliver excitations to the RC of FRL-PSII and power charge separation with a higher rate and efficiency than at least a fraction of the Chls *f* found in the FRL-PSII antenna (CP43 and CP47 paralogs). Furthermore, the excitations formed in the FRL-PBSs do not equilibrate entirely with the Chls *f* of FRL-PSII, not even after 2 ns. This behavior is different from most known photosynthetic systems/organisms and is, therefore, highly unexpected.

The lack of energy equilibration is usually ascribed to disrupted energetic connectivity between pigments. However, FRL-PBSs display variable fluorescence and their excitations are trapped in the RC within 500 ps, implying that they are energetically well connected to the RCs of FRL-PSII. The only way to account for these seemingly contrasting results is to assume that, for structural reasons, the energy transfer route from FRL- PBSs to the RC of FRL-PSII is faster than that connecting at least some of the

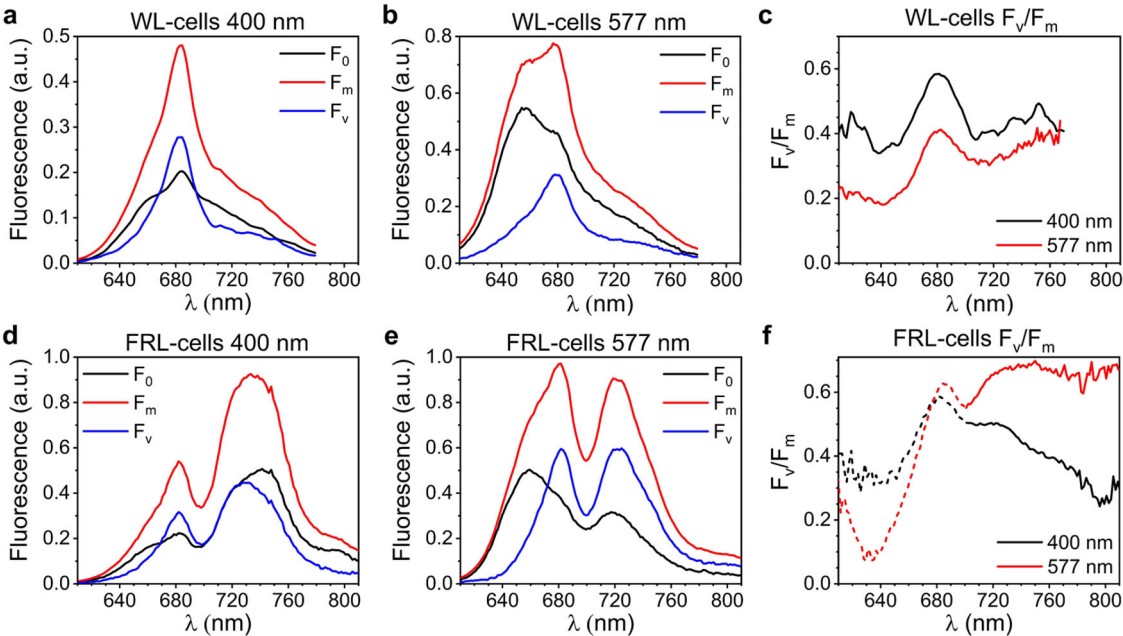

**Fig. 3 Variable fluorescence and $F_v/F_m$ of WL- and FRL-cells excited at different wavelengths at room temperature. a, b** Steady-state fluorescence spectra of WL-cells in open ($F_o$) and closed ($F_m$) state excited at 400 nm **a** and 577 nm **b** obtained from the DAS in Supplementary Figs. 4 and 5 and normalized to the area of the time-zero spectrum. Variable fluorescence spectra ($F_v$) are calculated as the difference between $F_m$ and $F_o$. **c** $F_v/F_m$ of WL-cells at different wavelengths calculated from the spectra in **a, b. d, e** Steady-state fluorescence spectra of FRL-cells in open ($F_o$) and closed ($F_m$) state excited at 400 nm **d** and 577 nm **e** obtained from the DAS in Supplementary Figs. 6 and 7. **f** $F_v/F_m$ of FRL-cells at different wavelengths calculated from the spectra in **d, e**. Note that the $F_v/F_m$ values around 680 nm upon 400-nm excitation in **f** are likely underestimated due to the presence of a small fraction of WL-PSII with closed RCs in the measurements at low powers (meaning that the integrated fluorescence in this region does not truly represent $F_o$; see the DAS in Supplementary Fig. 6). As a result, these $F_v/F_m$ values cannot be directly compared to those with 577-nm excitation in the same region, which is why they are shown in dashed lines. On the other hand, FRL-PSII RCs are fully open in the measurements performed at low power with both 400-nm and 577-nm excitations (see Supplementary Fig. 11 for details). Consequently, the integrated fluorescence data of FRL-cells above 700 nm are representative of $F_o$ and $F_m$, and the corresponding $F_v/F_m$ values (continuous lines) can be used to compare the efficiencies of trapping by FRL-PSII RCs upon either Chl or bilin excitation.

Chls $f$ in the antenna and the RC itself. This also explains why the excitations formed in the FRL- PBSs can reach the RC without equilibrating with all the Chls $f$ in the FRL-PSII antenna. The shortcut connecting the FRL- PBSs to the FRL-PSII RCs is likely to involve an intermediate Chl in the antenna (CP43/47 paralogs) acting as a bridge. Indeed, structural data on canonical PBSs have shown that the pigments in the APC core are too far from those of the PSII RC to allow direct energy transfer between them[8,9,33]. To allow for faster trapping of bilin excitations, this bridging Chl in the antenna needs to be better connected to the RC than most other red-shifted Chls and, to avoid energetic barriers when receiving excitations from FRL-PBSs, it is likely to be red-shifted as well.

This peculiar kinetics is the result of the small number of red-shifted pigments surrounded by a majority of Chls $a$. In light of this architecture, the connectivity between the FRL-PSII antenna and the RC can only count on the few Chls $f$ in CP43/47, which rapidly harvest all excitations from the surrounding Chls $a$[19], and the single red-shifted pigment in the RC, most likely the primary donor, $Chl_{D1}$[17]. The couplings between these Chls are relatively weak and, as a result, the overall energy migration towards the RC is slowed down relative to WL-PSII[19]. At the same time, the fluorescence decay of purified FRL-PSII in open state is, however, multi-exponential:[19] this can be interpreted in terms of the RCs trapping excitations from different red-shifted Chls in the antenna on different timescales, from 100–200 ps to about 1 ns.

The above view is confirmed by the structure of the FRL PSII core that was recently determined[20]. The structure is very similar to that of the canonical PSII core of *T. vulcanus* (Umena et al.[34].),

but five Chls $a$ have been replaced by four Chls $f$ and 1 Chl $d$ per monomeric core (see Fig. 4). $Chl_{D1}$, the primary electron donor in the RC, is identified as a Chl $d$, whereas CP43 contains only one Chl $f$ at position 507 and CP47 contains 3 Chls $f$ at positions 605, 608 and 614. Because all Chls in either CP43 or CP47 are well connected to each other regarding excitation energy transfer[19,35], excitations on Chls $a$ are rapidly transferred (within several ps) to the various Chls $f$ (Chl 507 in case of excitation in CP43 and Chls 605, 608 and 614 in CP47). Because the excitation energy of Chl $f$ is far lower than that of Chl $a$, hardly any back transfer towards Chl $a$ takes place. Chl 507 can transfer its excitation energy subsequently to the primary donor $Chl_{D1}$ in the RC whereas the Chls $f$ in CP47 can transfer the excitation energy either towards each other or to $Chl_{D1}$, from where charge separation takes place. Based on the crystal structure[20], we have estimated the excitation-energy transfer times from the four Chls $f$ towards $Chl_{D1}$ making use of the Förster equation. The results can be found in Supplementary Table 1 and the hopping times (inverse of energy transfer rates) are shown in Fig. 4. Although these numbers are only approximately correct, their relative values are quite accurate and the transfer from Chl 507 to Chl D1 in CP43 is much faster than that from the Chls in CP47, where in fact an average rate should be considered for the three Chls $f$.

The transfer time of 259 ps calculated from Chl 507 of CP43 to the Chl $d$ in the RC is relatively close to the 190 ps observed for FRL PSII cores[19]. Because CP47 contains three Chls $f$ we calculated their average transfer rate $k_{av}$, which is 1/3 $((0.45\,\text{ns})^{-1} + (0.99\,\text{ns})^{-1} + (1500\,\text{ns})^{-1}) = (0.93\,\text{ns})^{-1}$. The corresponding transfer time of 0.93 ns is rather close to the

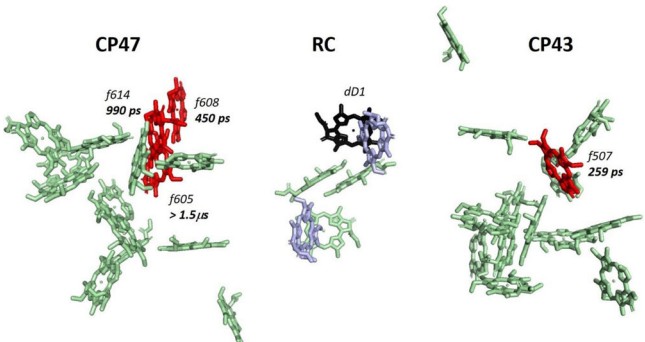

**Fig. 4 Hopping times of excitations from Chl *f* molecules in the PSII core antenna to Chl *d* in the reaction center.** The Chl binding sites are taken from the PSII structure of *Synechococcus* sp. PCC 7335 by Grisriel et al.[20]. CP43 contains 1 Chl *f* (pigment 507, numbering as in the PSII core of *T. vulcanus*[35]) that can transfer to Chl$_{D1}$. Three Chls *f* are coordinated by CP47. The hopping times (inverse of energy transfer rates) are taken from Supplementary Table 1 and are calculated under the assumption that all Chls *f* emit at 740 nm (based on the average emission wavelength of FRL-PSII) and the acceptor (Chl$_{D1}$) absorbs at 727 nm (based on Nurnberg et al. and Gisriel et al.)[17,20]. Chl$_{D1}$ is shown in black, while the Chls *f* are in red. The two pheophytin molecules in the RC are displayed in light blue, all other Chls are in light green.

experimentally determined value of 1.0 ns. In light of the structure of the core a straightforward interpretation is that the 1.0 ns component reflects transfer from CP47 to Chl$_{D1}$ followed by charge separation, and the 190 ps component represents transfer from CP43 to Chl$_{D1}$. Because CP47 contains more Chls than CP43 (presumably 16 vs. 13 Chls) one would expect a somewhat higher amplitude for the 1.0 ns component, which was indeed observed.

As was already mentioned in the Introduction, the exact composition of the PBSs in the FRL-acclimated cells used in this study is not known, but it has been recently found that they are smaller than the canonical PBSs found in WL-acclimated cells[27]. For the ease of discussing our results, we assume that the FRL-PBSs consist simply of FRL-BCs like in the structural paper of Gisriel et al.[20] and demonstrate that this can explain our results in a straightforward way. According to the structural model provided by Gisriel et al.[20], there is a relatively close connection between bilins in the FRL-bicylindrical cores and Chl 507 in CP43, whereas a good connection with CP47 is missing. If there would have been a good connection with CP47 this would have led to a transfer time from FRL-BC's to Chl$_{D1}$ well above 1.0 ns, i.e. the time of transfer from CP47 to Chl$_{D1}$. Instead, a fluorescence lifetime of 532 ps is observed upon 577-nm (FRL-PBSs) excitation, reflecting transfer to CP43, followed by transfer to Chl$_{D1}$. Because this is the longest time observed for open PSII RCs, this means that excitation energy transfer to CP47 is negligible. Only when the Chls are directly excited at 400 nm, a long fluorescence lifetime of 926 ps is observed, reminiscent of the 1.0 ns that is observed for isolated PSII cores[19]. Upon 400-nm excitation, a lifetime of 196 ps is also found, comparable to the 190 ps lifetime observed in isolated PSII cores[19] (transfer from CP43), but the corresponding DAS observed in the FRL-cells studied here also shows contributions from WL-PSII and FRL-PSI.

Finally, the weak connectivity between the Chls of CP43 and those of CP47 can account for the dependency of the fluorescence spectra on the excitation wavelength. Indeed, the excitations formed on FRL-PBSs can only equilibrate with the Chls of CP43 without reaching those of CP47, and vice versa.

As a result of the unusual architecture and energetics, FRL-BCs increase both the antenna size and the photochemical yield of FRL-PSII RCs simultaneously, which represents a unique case in photosynthetic systems. Based on the absorption spectra of the isolated complexes, it can be estimated that FRL-PBSs contain about twice as many FRL-absorbing pigments (~20 bilins; see Supplementary Fig. 12 for further details) as the FRL-PSII dimer (which binds ~10 Chls *f/d*)[17–20]. As a result, they increase the antenna size of FRL-PSII under FRL by a factor of 3. At the same time, FRL-PBS excitations are processed 1.5 times more efficiently than Chl excitations (on average, see previous section), implying that FRL-PBSs could increase the overall performance of FRL-PSII RCs by a factor of 4.5 in FRL. This finding can also shed light on some other peculiar features of FaRLiP. Indeed, while most photoacclimation strategies developed by photosynthetic organisms are restricted to the antenna, FaRLiP also remodels the photosystem cores, which change their protein and pigment composition[14]. The insertion of red-shifted pigments is not particularly advantageous in terms of PSII efficiency[19], but might be needed to ensure energetic connectivity between the FRL-PBS antenna, which contains the majority of FRL-absorbing pigments, and the RC. Indeed, a Chl *a*-only PSII core would hardly trap excitations from such a red-shifted antenna. Notably, FaRLiP is also substantially different from the strategy adopted by the only other known cyanobacterium able to grow in FRL, *Acaryochloris marina*[36]. This species also uses the red-shifted Chl *d* but, unlike FaRLiP strains, where Chl *a* remains the most abundant pigment also in FRL, *Acaryochloris* photosystems bind mostly Chl *d*[36–38]. The almost exclusive presence of Chl *d* results in a better energetic connectivity in the PSII core of *Acaryochloris*, which shows therefore higher photochemical yields than FRL-PSII of FaRLiP strains[19]. At the same time, however, the phycobiliproteins produced by *Acaryochloris* only absorb visible light[39], whereas FaRLiP strains harvest a substantial amount of FRL via red-shifted APC. In this view, the positioning of the long-wavelength Chls might be optimized for FRL-PSII to work in combination with FRL-PBSs rather than on its own. In addition, the observation that the antenna compensates for the relatively low efficiency of FRL-PSII alone might also partially explain why the latter does not accumulate in FaRLiP organisms depleted in FRL-absorbing APC[40].

Why do FaRLiP strains and *Acaryochloris* adopt such different strategies to convert low-energy photons? One possible explanation is that FaRLiP is an acclimative process allowing cyanobacteria to harvest FRL only when needed, whereas *Acaryochloris* employs Chl *d* permanently. The necessity to switch back to the more efficient Chl *a*-only photosystems when visible light becomes available might explain why FaRLiP cyanobacteria use Chl *a* as their major Chl also in FRL. As a result, the insertion of only few red-shifted Chls in the PSII core while most far-red photons are collected by the outer antenna would make FaRLiP cyanobacteria more flexible than *Acaryochloris* towards changes in the light spectrum. This hypothesis is also in line with the fact that FaRLiP has been adopted by many distantly-related species populating a variety of terrestrial and marine environments, whereas the strategy adopted by *Acaryochloris* remains unique[16,41]. Finally, the use of red-shifted Chls allows harvesting less energetic photons but also introduces additional challenges for the photosystems (such as a lower efficiency, and/or a higher probability of charge recombination)[17,19], which is why it is restricted to environments enriched in FRL.

## Methods
**Cell cultures**. The strain *Chroococcidiopsis thermalis* PCC 7203 was obtained from the Pasteur Culture Collection (Institut Pasteur, Paris, France) and grown at 30 °C in BG11[42] medium with addition of 20 mM HEPES-NaOH (pH = 8.0). Cells were grown

under white light (WL) of 30 $\mu$mol photons m$^{-2}$ s$^{-1}$ starting from OD$_{750}$ = 0.1 and collected for experiments at OD$_{750}$ = 0.8–1.0. Cells were also grown under far-red light (FRL, 738 nm) of 45 $\mu$mol photons m$^{-2}$ s$^{-1}$ for 2.5 to 3 weeks prior to experiments, starting from OD$_{750}$ = 0.3–0.4, with the medium being refreshed every week keeping an OD$_{750}$ = 1.0–1.5. The cultures were grown in Erlenmeyer flasks shaken at 100 rpm. All measurements were performed on intact cells in their logarithmic growth phase.

**Chlorophyll content determination.** Pigments from FRL-cells were extracted in pure methanol and their concentrations in the extract were determined by absorbance measurements. To improve the extraction efficiency of methanol, glass beads (about 30% v/v) were added and the mixture was vigorously shaken for 30 min in darkness. The absorption spectra of the Chls in the Q$_y$ region were fitted using the available spectra and extinction coefficients of the pure pigments in methanol[43]. Average relative concentrations (and standard deviations) of the Chls were obtained from 4 measurements on 2 distinct cell batches. Note that the presence of WL-photosystems (particularly WL-PSII) in FRL-cells has an influence on the overall Chl content.

**Steady-state spectroscopy.** Room-temperature (RT) absorption spectra of cells were acquired on a Varian Cary 4000 UV–Vis-spectrophotometer (Agilent technologies) equipped with an integrating diffuse reflectance sphere (DRA-CA-50, Labsphere) to correct for light scattering. RT fluorescence spectra were acquired at an OD < 0.05 cm$^{-1}$ at Q$_y$ maximum on a HORIBA Jobin-Yvon FluoroLog-3 spectrofluorometer. Absorption and fluorescence spectra were measured on at least three biological replicates for both FRL- and WL- acclimated cells, yielding similar results.

**Time-resolved fluorescence.** Time-resolved fluorescence (TRF) measurements with 400-nm excitation were performed with a synchro scan streak camera setup[19]. TRF measurements with 577-nm excitation were recorded with a slightly different synchro scan streak camera setup[44,45]. In brief, the laser repetition rate was 3.8 MHz and the time window 2.0 ns. 577-nm excitation measurements with nearly all PSII RCs open were performed with an excitation power of 0.2 $\mu$W (with a sample volume of 2 mL), while those with closed PSII RCs were performed at 2 $\mu$W, after addition of 3-(3,4-dichlorophenyl)-1,1-dimethylurea (DCMU) and pre-illumination with white light for one minute (with a sample volume of 1 mL). The intensity dependent measurements are shown in Supplementary Fig. 11. The experiments were performed at RT in a magnetically stirred 1 cm × 1 cm cuvette with a sample OD of about 0.5 cm$^{-1}$ at Q$_y$ maximum (excitation and detection from the sample at the edge of the cuvette, thereby avoiding self-absorption) and a measuring time from 15 minutes to 1 hour of CCD exposure (no sample degradation was observed during the measuring time). The averaged images were corrected for background and shading, and then sliced into traces of ~1.5-nm width prior to analysis. 400-nm excitation experiments consisted of three (two) biological replicas for FRL- (WL-) adapted cells, while 577-nm experiments consisted of two (one) biological replicas for FRL- (WL-) adapted cells. The biological replicas yielded very similar results. All TRF experiments were performed at RT.

**Data analysis.** Fluorescence time traces were globally analyzed with Glotaran and the TIMP package for R[46] using a number of parallel kinetic components. The total dataset can be described by the fitting function $f(t, \lambda)$:

$$f(t, \lambda) = \sum_{k=1}^{n} DAS_k(\lambda) \cdot \exp\left(-\frac{t}{\tau_k}\right) \otimes IRF(t, \lambda)$$

where each decay-associated spectrum ($DAS_k$) is the amplitude factor associated with a decay component $k$ having a decay lifetime $\tau_k$. The instrument response function $IRF(t, \lambda)$ was estimated from the fitting (FWHM ~ 20 ps) using a Gaussian profile. The fitting also accounts for time-zero dispersion. In some cases, a double Gaussian was required, consisting of a ~20 ps FWHM (90% of IRF area) on top of a Gaussian of ~100 ps FWHM (10% of IRF area). For each experiment, the time-zero spectrum was obtained by summing all DAS. The steady-state fluorescence spectra were reconstructed by integrating the TRF data as:

$$F(\lambda) = \sum_{k=1}^{n} DAS_k(\lambda) \cdot \tau_k$$

In order to compare results from different measurements, the DAS and reconstructed steady-state fluorescence spectra were normalized to the area of the time-zero spectrum (corresponding to the same amount of initial excitation) in each dataset. Overlays of the raw and fitted kinetic traces can be found in Supplementary Fig. 9.

**Reporting summary.** Further information on research design is available in the Nature Research Reporting Summary linked to this article.

## Data availability

All data needed to evaluate the conclusions in this paper are present in the paper and/or Supplementary Information. Source data files are available for all figures in the manuscript. All original data are available from the authors upon reasonable request. Source data are provided with this paper.

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

## Acknowledgements

This project was supported by by the Netherlands Organization for Scientific Research (NWO) via a Top grant to R.C. and the European Union's Horizon 2020 research and innovation program under the Marie Skłodowska-Curie grant agreement no. 675006 (to H.v.A. and R.C.). L.B. was supported by an EMBO long-term fellowship (EMBO ALTF 292-2017).

## Author contributions

V.M, L.B., and R.C. conceived the research. V.M., A.F.B., and L.B. performed the experiments. V.M., H.v.A., and R.C. analyzed the data; V.M. wrote the paper with contributions from H.v.A. and R.C.

## Competing interests

The authors declare no competing interests.
