## [Peer Review File · Nature Communications]

The antenna of far-red absorbing cyanobacteria increases both absorption and quantum efficiency of Photosystem IIREVIEWER COMMENTS

Reviewer #1 (Remarks to the Author):

The work by Mascoli et al. is devoted to quantitative analysis of photochemical reactions taking place in cells of cyanobacterium *Chroococciopsis thermalis* PCC 7203 adopted for visible and far-red light conditions. However, the work contains some methodological flaws that make quantitative analysis implausible. In particular, the paper compares kinetic measurements performed on whole cells grown in white light and under infrared light conditions. In the latter case, the cells contained a mixture of complexes FRL-PSI, FRL-PSII, WL-PSI, WL-PSII (lines 67-69), however, their relative concentrations were not determined. The conclusions, which were drawn on the basis of a qualitative comparison of the fluorescence kinetics obtained for cells with different pigment compositions under different excitation conditions (400 nm and 577 nm), were not verified using quantitative kinetic modeling. However, the crucial methodological fault of this work lies in the incorrect assumption that the fluorescence of cells with "closed" PSII complexes is not associated with PSII, which, according to the authors, makes it possible "to separate the spectroscopic contribution of PSII from that of PSI and estimate the photochemical yield of PSII" (line 117).

Pre-illumination of PSII with white light in the presence of DCMU (competitive inhibitor of the secondary quinone QB) leads to pre-reduction of the primary quinone QA; under these conditions, photoinduced charge separation is restricted by the formation of ion-radical state $P680(+)\text{HA}(-)\text{QA}(-)$ [Roelofs T. A., Lee C. H. & Holzwarth A. R. (1992) Global target analysis of picosecond chlorophyll fluorescence kinetics from pea chloroplasts: a new approach to the characterization of the primary processes in photosystem II α - and β -units. *Biophysical Journal*, 61(5), 1147-1163]. The recombination of $P680(+)\text{HA}(-)\text{QA}(-)$ proceeds at ~ 4 ns [Shuvalov V. A., Klimov V. V., Dolan E., Parson W. W. & Ke B. (1980) Nanosecond fluorescence and absorbance changes in photosystem II at low redox potential: Pheophytin as an intermediary electron acceptor. *Febs Letters*, 118(2), 279-282] and the large increase of chlorophyll fluorescence upon the QA reduction is due to a decrease in competition between trapping and emission of PSII [Barber J., Malkin S. & Telfer A. (1989) The origin of chlorophyll fluorescence in vivo and its quenching by the photosystem II reaction centre. *Philosophical Transactions of the Royal Society of London. B, Biological Sciences*, 323(1216), 227-239]. For this reason, the increased fluorescence of cells with "closed" PSII complexes is due to the fluorescence of the inhibited PSII itself; therefore, the use of "closed" PSII complexes does not in any way allow one to determine the contributions of PSI and PSII to the fluorescence quenching kinetics.

"Efficiency of PSII photochemistry upon different pigment excitations." (lines 176-181)

The main results of this work were obtained by subtraction of fluorescence spectra in the open (F_o) and closed (F_m) states with a goal to calculate the variable fluorescence ($F_v = F_m - F_o$). Because the fluorescence in the open (F_o) and closed (F_m) states at large delays originates both from the FRL-PSII complexes, the difference $F_m - F_o$ can't be a measure of the efficiency of the PSII photochemistry, and the results made on the basis of such an analysis are not reliable.

Reviewer #2 (Remarks to the Author):

The paper raises a very interesting new idea, that the efficiency of light-harvesting in a far-red acclimated cyanobacterium can be boosted by a selective fast-track for energy transfer from long-wavelength phycobilins to a subset of long-wavelength chlorophylls in close contact with the special pair of the reaction centre. I think the idea is plausible and could give nice insights into how cyanobacteria manage to utilise far-red light at the edge of what is theoretically possible. There could also be implications for the design of bio-inspired artificial devices. However, I don't find the data completely convincing as it stands. The problem is that the crucial comparisons (between WL and FRL cells, between 400 and 577 nm excitation etc) are extremely complex, and I am not sure that we have

enough background information to be able to interpret the data reliably. I don't think the data and the arguments are presented as clearly and sharply as they could be: I found it took a big effort to try to work my way through this. So it may be that I've got confused and missed something, but in any case the authors need to do some significant work on this to make a clear and convincing story.

Specific points:

1. Lines 90-96. It's frustrating that we don't have more information on what is happening with the PBS formed under FRL acclimation in this particular species (*C. thermalis*). The authors talk about FRL-BCs (ie just PBS cores that include red-shifted chromophores) but they comment in the Introduction that such FRL-BCs can bind peripheral PBS rods, or not, in other species (lines 64-67). It doesn't seem clear whether or not this is the case in *C. thermalis*: the cited references don't discuss this species as far as I can see. If such peripheral rods are there, they will be preferentially excited with the 577 nm light used throughout, and so this will make a difference to the overall kinetics. I don't think the spectroscopic information (Fig. 1, Fig. S7) can give a clear-cut answer because the FRL-adapted cells still contain a proportion of WL PBS (line 121). If the FRL PBS are much smaller than the WL PBS, then it really complicates kinetic comparisons. Faster overall kinetics with the FRL PBS might be more a consequence of their small size rather than fast-tracking of the excitons through long-wavelength chlorophylls. I suspect that some comparison of isolated WL and FRL PBS from *C. thermalis* may be necessary to provide enough background information to interpret the in vivo data.

2. Fig. 2B and associated text on pages 4-5. The faster 730 nm fluorescence decay (at 0.3 ns and beyond) with 577 nm rather than with 400 nm excitation might represent the simplest and most direct evidence for particularly efficient utilisation of FRL-BC-absorbed light. However, the interpretation depends critically on where the fluorescence is coming from. The authors choose to normalise and present the traces from 0.3 ns onwards to avoid issues with the fast-decaying PSI fluorescence (lines 138-140). But PSI isn't the only problem. There is also the issue with overlap with phycobilin fluorescence, where, for example, there is a 161 ps component still with significant amplitude (similar to that of PSII) at 730 nm (Fig. S7A). The decay is faster than PSII, but not so fast that you can exclude it by looking only from 300 ps onwards. So the simplest explanation for the kinetic differences seen in Fig. 2B is that the trace with 577 nm excitation includes a faster PBS component that is not seen with 400 nm excitation. That possibility really has to be rigorously excluded before you can make any conclusions about the efficiency of PSII energy trapping. Please note here and elsewhere that phycobilins always make unexpectedly large contributions to DAS amplitudes as compared to chlorophylls because their radiative lifetimes are substantially shorter (ie the intrinsic efficiency of fluorescence is greater).

3. Fig. 2C and associated text on p.5. The PSII 532 ps DAS has two peaks in it, at about 680 nm and 730 nm (Fig. 2C). This doesn't seem to be discussed, but I guess it is what you would expect: the 680 nm peak comes from the remaining WL-PSII and the 730 nm peak comes from the FRL-PSII. The striking thing is that the lifetimes seem to be similar enough for both peaks to appear in the same DAS! Indeed, the equivalent component in WL cells is 503 ns (Fig S5a), which does not look significantly different to the 532 ps resolved in FRL cells. At first sight, that's hard to reconcile with authors' interpretation that the lifetime shows the overall kinetics of PSII energy trapping (lines 144-145), and that the process is particularly efficient in FRL complexes. Is the explanation that faster light-harvesting with FRL-BCs is cancelled out by intrinsically slower charge-separation in FRL-PSII? If so, that needs to be discussed and justified more carefully. I still worry whether the faster light-harvesting might be a simple consequence of the smaller size of the FRL-BCs as compared to the WL-PBS.

4. Fig. 3. I think the use of reconstructed Fv/Fm ratios in this figure just causes confusion, and I would strongly recommend removing the figure altogether. I think the answers should be in the fluorescence decay kinetics and the DAS, not in the Fv/Fm ratios. "Fv/Fm" in these measurements is clearly not any sort of measure of the quantum yield of PSII photochemistry as suggested in line 181. The values

seen are far too low for that. There are multiple problems with Fv/Fm measurement in cyanobacteria. One problem is from PSI fluorescence, and that is explored a bit in Fig. S10 (see line 181). But then there is also the more serious issue of overlap of PSII chl fluorescence with phycobilin fluorescence, which is alluded to in lines 201-21 but not fully discussed. If the excitons are not in equilibrium (see eg lines 104-105) then phycobilin fluorescence will not respond to PSII trap closure as strongly as chl fluorescence, so phycobilin fluorescence contributes much more to Fo than to Fv. Furthermore, phycobilins contribute disproportionately to the overall fluorescence because of the shorter radiative lifetime (see 2. above). Under those conditions, Fv/Fm doesn't measure quantum efficiency, and of course there is a particular problem if you want to compare Fv/Fm ratios across the emission spectrum and with chlorophyll and phycobilin excitation as in lines 201-227. That's really why you need the time-resolved decays and the DAS!

Response to the reviewers

Reviewer #1 (Remarks to the Author):

Comment: The work by Mascoli et al. is devoted to quantitative analysis of photochemical reactions taking place in cells of cyanobacterium *Chroococcidiopsis thermalis* PCC 7203 adopted for visible and far-red light conditions. However, the work contains some methodological flaws that make quantitative analysis implausible. In particular, the paper compares kinetic measurements performed on whole cells grown in white light and under infrared light conditions. In the latter case, the cells contained a mixture of complexes FRL-PSI, FRL-PSII, WL-PSI, WL-PSII (lines 67-69), however, their relative concentrations were not determined. The conclusions, which were drawn on the basis of a qualitative comparison of the fluorescence kinetics obtained for cells with different pigment compositions under different excitation conditions (400 nm and 577 nm), were not verified using quantitative kinetic modeling.

Answer: Indeed we did not use quantitative kinetic modeling, which is according to us not possible without making strong assumptions. However, we show that we can separate the kinetics of the WL and the FRL components and we draw conclusions based on solid qualitative comparisons.

Continuation of the comment: However, the crucial methodological fault of this work lies in the incorrect assumption that the fluorescence of cells with "closed" PSII complexes is not associated with PSII, which, according to the authors, makes it possible "to separate the spectroscopic contribution of PSII from that of PSI and estimate the photochemical yield of PSII" (line 117).

Answer: We do not claim anywhere that the fluorescence of cells with "closed" PSII is not associated with PSII. On the contrary, closing PSII leads to strongly increased PSII fluorescence while PSI fluorescence remains unaltered. This is now explicitly mentioned in for example the paragraph "Trapping of Chl and bilin excitations by FRL-PSII RCs" in the Results.

Continuation of the comment: Pre-illumination of PSII with white light in the presence of DCMU (competitive inhibitor of the secondary quinone QB) leads to pre-reduction of the primary quinone QA; under these conditions, photoinduced charge separation is restricted by the formation of ion-radical state P680(+)/HA(-)/QA(-) [Roelofs T. A., Lee C. H. & Holzwarth A. R. (1992)Biophysical Journal, 61(5), 1147-1163]. The recombination of P680(+)/HA(-)/QA(-) proceeds at ~4 ns [Shuvalov V. A., Klimov V. V., Dolan E., Parson W. W. & Ke B. (1980)Febs Letters, 118(2), 279-282] and the large increase of chlorophyll fluorescence upon the QA reduction is due to a decrease in competition between trapping and emission of PSII [Barber J., Malkin S. & Telfer A. (1989)B, Biological Sciences, 323(1216), 227-239]. For this reason, the increased fluorescence of cells with "closed" PSII complexes is due to the fluorescence of the inhibited PSII itself; therefore, the use of "closed" PSII complexes does not in any way allow one to determine the contributions of PSI and PSII to the fluorescence quenching kinetics.

Answer: indeed, closing PSII leads to strongly increased PSII fluorescence (because of the longer fluorescence lifetimes). Because PSI fluorescence remains unaltered, the variable fluorescence is due to PSII and the invariable part is due to PSI. This allows separation of both contributions.

Comment: "Efficiency of PSII photochemistry upon different pigment excitations." (lines 176-181)
The main results of this work were obtained by subtraction of fluorescence spectra in the open (F_o) and closed (F_m) states with a goal to calculate the variable fluorescence ($F_v = F_m - F_o$). Because the fluorescence in the open (F_o) and closed (F_m) states at large delays originates both from the FRL-PSII

complexes, the difference Fm-Fo can't be a measure of the efficiency of the PSII photochemistry, and the results made on the basis of such an analysis are not reliable.

Answer: We cannot follow the reviewer here. We fully agree that the fluorescence in the open (Fo) and closed (Fm) states at large delays originates both from the FRL-PSII complexes. That is why Fv/Fm reflects the efficiency of these FRL-PSII complexes.

Reviewer #2 (Remarks to the Author):

Comment: The paper raises a very interesting new idea, that the efficiency of light-harvesting in a far-red acclimated cyanobacterium can be boosted by a selective fast-track for energy transfer from long-wavelength phycobilins to a subset of long-wavelength chlorophylls in close contact with the special pair of the reaction centre. I think the idea is plausible and could give nice insights into how cyanobacteria manage to utilise far-red light at the edge of what is theoretically possible. There could also be implications for the design of bio-inspired artificial devices. However, I don't find the data completely convincing as it stands. The problem is that the crucial comparisons (between WL and FRL cells, between 400 and 577 nm excitation etc) are extremely complex, and I am not sure that we have enough background information to be able to interpret the data reliably. I don't think the data and the arguments are presented as clearly and sharply as they could be: I found it took a big effort to try to work my way through this. So it may be that I've got confused and missed something, but in any case the authors need to do some significant work on this to make a clear and convincing story.

Answer: We thank the reviewer for their encouraging words and their thorough study of the manuscript; we realize that it may well have been a tour de force to work their way through it. It is a complex system to study and analyze but also to report about the results in a concise and clear way. We are fortunate that very recently (after writing and submitting our previous version) the structure of the FarLiP PSII core was published by Don Bryant and coworkers. This structure is in very nice agreement with the conclusions that we have drawn from the time-resolved work in our study. It has allowed us now to rewrite the Discussion of the manuscript in a way that is hopefully far easier to understand.

Comment:

Specific points:

1. Lines 90-96. It's frustrating that we don't have more information on what is happening with the PBS formed under FRL acclimation in this particular species (*C. thermalis*). The authors talk about FRL-BCs (ie just PBS cores that include red-shifted chromophores) but they comment in the Introduction that such FRL-BCs can bind peripheral PBS rods, or not, in other species (lines 64-67). It doesn't seem clear whether or not this is the case in *C. thermalis*: the cited references don't discuss this species as far as I can see. If such peripheral rods are there, they will be preferentially excited with the 577 nm light used throughout, and so this will make a difference to the overall kinetics. I don't think the spectroscopic information (Fig. 1, Fig. S7) can give a clear-cut answer because the FRL-adapted cells still contain a proportion of WL PBS (line 121). If the FRL PBS are much smaller than the WL PBS, then it really complicates kinetic comparisons. Faster overall kinetics with the FRL PBS might be more a consequence of their small size rather than fast-tracking of the excitons through long-wavelength chlorophylls. I suspect that some comparison of isolated WL and FRL PBS

from *C. thermalis* may be necessary to provide enough background information to interpret the *in vivo* data.

Answer: It is indeed not known whether the antenna consists of FRL-BCs only or whether there are also some peripheral PBS rods present. However, this information is not needed for the conclusions that we are drawing. The important part is that trapping of excitations in the presence of this antenna is faster than trapping in the absence of the antenna. We now demonstrate that our results nicely agree with the new structural study of Bryant and coworkers but the overall reasoning that we assume remains identical in case the antenna would also include additional rods.

Comment: 2. Fig. 2B and associated text on pages 4-5. The faster 730 nm fluorescence decay (at 0.3 ns and beyond) with 577 nm rather than with 400 nm excitation might represent the simplest and most direct evidence for particularly efficient utilisation of FRL-BC-absorbed light. However, the interpretation depends critically on where the fluorescence is coming from. The authors choose to normalise and present the traces from 0.3 ns onwards to avoid issues with the fast-decaying PSI fluorescence (lines 138-140). But PSI isn't the only problem. There is also the issue with overlap with phycobilin fluorescence, where, for example, there is a 161 ps component still with significant amplitude (similar to that of PSII) at 730 nm (Fig. S7A). The decay is faster than PSII, but not so fast that you can exclude it by looking only from 300 ps onwards.

Answer: Indeed there is also fluorescence from -PBSs after 0.3 ns and this also contributes to the faster decay but mainly at early times after $t=0.3$ ns. Nevertheless, it can still be concluded that the trapping is faster in FRL-PSII for two reasons: 1) also far beyond 0.3 ns the decay is still slower for chlorophyll excitation as can be seen from the slope lines in figure 2B of the manuscript, and 2) the slowest component (see DAS in figure 2C) corresponds to a lifetime of 532 ps for PBS excitation at 577 nm and a (far) slower lifetime of 926 ps upon Chl excitation at 400 nm. We have reformulated the text on pages 4-5 to make this more explicit. In view of the new structural information the difference can easily be explained by the fact that PBS excitation leads to energy transfer via CP43 only which is well connected to the RC whereas at 400 nm also CP47 is excited, which is far less well connected to the RC.

Continuation comment: So the simplest explanation for the kinetic differences seen in Fig. 2B is that the trace with 577 nm excitation includes a faster PBS component that is not seen with 400 nm excitation. That possibility really has to be rigorously excluded before you can make any conclusions about the efficiency of PSII energy trapping. Please note here and elsewhere that phycobilins always make unexpectedly large contributions to DAS amplitudes as compared to chlorophylls because their radiative lifetimes are substantially shorter (ie the intrinsic efficiency of fluorescence is greater).

Answer: We agree that the trace with 577 nm excitation includes a fast PBS component (that reaches the RC via CP43) and this is in fact our explanation for the obtained results. It is probably important to note that when we speak about the efficiency of energy trapping that this includes the effect of the antenna. Hopefully this is now also more clear from the text. In relation to this it is important to emphasize that we agree with the reviewer that the radiative rates of phycobilins are higher than those of chlorophylls but on the other hand also Chl *f* has an oscillator strength of about 1.35 times that of Chl *a*. This means that transfer from PBS to PSII core does not lead to a decay of the fluorescence but mainly to a shift of the emission maximum. Hopefully this clarifies the approach and interpretation of the data that we used.

Comment: 3. Fig. 2C and associated text on p.5. The PSII 532 ps DAS has two peaks in it, at about 680 nm and 730 nm (Fig. 2C). This doesn't seem to be discussed, but I guess it is what you would expect: the 680 nm peak comes from the remaining WL-PSII and the 730 nm peak comes from the FRL-PSII. The striking thing is that the lifetimes seem to be similar enough for both peaks to appear in the same DAS! Indeed, the equivalent component in WL cells is 503 nm (Fig S5a), which does not look significantly different to the 532 ps resolved in FRL cells. At first sight, that's hard to reconcile with authors' interpretation that the lifetime shows the overall kinetics of PSII energy trapping (lines 144-145), and that the process is particularly efficient in FRL complexes. Is the explanation that faster light-harvesting with FRL-BCs is cancelled out by intrinsically slower charge-separation in FRL-PSII? If so, that needs to be discussed and justified more carefully. I still worry whether the faster light-harvesting might be a simple consequence of the smaller size of the FRL-BCs as compared to the WL-PBS.

Answer: This topic was indeed not discussed in the associated text on p.5 but in the supplementary material (Fig S6). and we now explicitly refer to it. Indeed the 680 nm peak of the 532 ps DAS is due to WL cells (503 ps in Fig S5a), while the 730 nm peak stems from FRL-PSII. The fact that the lifetime is so similar for both systems indicates that the FRL-BC is smaller as compared to the WL-PBS. We have now explicitly mentioned that in the text, just above the paragraph "Efficiency of PSII photochemistry ...", we agree with the reviewer that this is indeed an important point. For us this is not worrisome, it is even the most likely explanation. For us the most striking observation is that this lifetime is far shorter than the lifetime for FRL PSII core without antenna (our Nature Plants paper) or of directly excited FRL PSII core with antenna (this work).

Comment: 4. Fig. 3. I think the use of reconstructed Fv/Fm ratios in this figure just causes confusion, and I would strongly recommend removing the figure altogether. I think the answers should be in the fluorescence decay kinetics and the DAS, not in the Fv/Fm ratios. "Fv/Fm" in these measurements is clearly not any sort of measure of the quantum yield of PSII photochemistry as suggested in line 181. The values seen are far too low for that. There are multiple problems with Fv/Fm measurement in cyanobacteria. One problem is from PSI fluorescence, and that is explored a bit in Fig. S10 (see line 181). But then there is also the more serious issue of overlap of PSII chl fluorescence with phycobilin fluorescence, which is alluded to in lines 201-21 but not fully discussed. If the excitons are not in equilibrium (see eg lines 104-105) then phycobilin fluorescence will not respond to PSII trap closure as strongly as chl fluorescence, so phycobilin fluorescence contributes much more to Fo than to Fv. Furthermore, phycobilins contribute disproportionately to the overall fluorescence because of the shorter radiative lifetime (see 2. above). Under those conditions, Fv/Fm doesn't measure quantum efficiency, and of course there is a particular problem if you want to compare Fv/Fm ratios across the emission spectrum and with chlorophyll and phycobilin excitation as in lines 201-227. That's really why you need the time-resolved decays and the DAS!

Answer: It is unfortunate that this figure caused confusion with the reviewer, but because we believe that it can be very informative, provided that we explain it clearly, we prefer to keep it in. We respectfully disagree with the two main criticisms that the reviewer raises and we would like to explain why. Also PSI has some influence on the data but this is less important and does not influence the overall conclusions

- 1- As was also mentioned above, Chl *f* has an oscillator strength that is about 1.35 times that of Chl *a*, and therefore also a shorter radiative lifetime than Chl *a*, somewhat comparable to

that of the phycobilins, so this will not cause a serious disproportionality due to differences in fluorescence lifetimes.

- 2- Indeed the excitons are not equilibrated over the phycobilins and the chlorophylls and therefore they respond less to the closure of the RCs but it simply means that the efficiency becomes lower when PBSs are excited instead of Chls. Usually F_v/F_m is used as a proxy for PSII efficiency and it is commonly explained by assuming that charge separation in the RCs is trap-limited. However, the approximation is equally valid if charge separation is migration-limited, which is at least partly the case upon PBS excitation, leading to a relatively large phycobilin contribution to the overall fluorescence, especially in the case of open RCs. We would like to emphasize that the PSII efficiency in our case is not simply the intrinsic efficiency of the PSII core but it includes the whole system, PSII core plus PBS, and the efficiency becomes wavelength dependent. We have tried to explain this more clearly in the current text. We believe that with this background information it is easier to interpret the F_v/F_m spectra and to compare them for WL and FRL cells. At least for us it turned to be very insightful. We agree that all the information is also in the time-resolved decays and the DAS but they provide an alternative visualization of the same effects and because we present both, the reader can choose their favorite approach.

Finally, we would also like to refer to Figures S10C-D, from which it is very clear in our opinion that F_v/F_m is always larger upon 577 nm excitation (antenna) than upon 400 nm excitation (core), even for the extreme ways of calculating (upper and lower limits) the values of F_v/F_m .

REVIEWERS' COMMENTS

Reviewer #2 (Remarks to the Author):

Overall I think this is a good response to my comments on the first version. The discussion is now much easier to read, and the incorporation of discussion around the new structure for FR-PSII makes the findings more concrete. I am still sceptical about the usefulness of the reconstructed Fv/Fm values. The authors comment in their rebuttal that chl f will have a radiative lifetime "somewhat comparable to that of the phycobilins". In fact I think it must be shorter than for chl a but still substantially longer than for the bilins, therefore Fv/Fm cannot provide a quantitative measure of trapping efficiency. However, I don't think this point invalidates any of the authors' main arguments.